# Insights into the computer-aided drug design and discovery based on anthraquinone scaffold for cancer treatment: A protocol for systematic review

**Hui Ming Chua** [1]*, **Said Moshawih**[1], **Hui Poh Goh**[1], **Long Chiau Ming**[2], **Nurolaini Kifli**[1]

**1** PAP Rashidah Sa'adatul Bolkiah Institute of Health Sciences, University Brunei Darussalam, Gadong, Brunei Darussalam, **2** School of Medical and Lifesciences, Sunway University, Bandar Sunway, Malaysia

* hmchua.pharm@gmail.com

**Data Availability Statement:** No datasets were generated or analysed during the current study.

**Funding:** The author(s) received no specific funding for this work.

## Abstract

There is still unmet medical need in cancer treatment mainly due to drug resistance and adverse drug events. Therefore, the search for better drugs is essential. Computer-aided drug design (CADD) and discovery tools are useful to streamline the lengthy and costly drug development process. Anthraquinones are a group of naturally occurring compounds with unique scaffold that exert various biological properties including anticancer activities. This protocol describes a systematic review that provide insights into the computer-aided drug design and discovery based on anthraquinone scaffold for cancer treatment. It was prepared in accordance with the "Preferred reporting items for systematic review and meta-analysis protocols (PRISMA-P) 2015 guidelines, and published in the "International prospective register of systematic reviews" database (PROSPERO: CRD42023432904). Search strategies will be developed based on the combination of relevant keywords and executed in PubMed, Scopus, Web of Science and MedRxiv. Only original studies that employed CADD as primary tool in virtual screening for the purpose of designing or discovering anti-cancer drugs involving anthraquinone scaffold published in English language will be included. Two independent reviewers will be involved to screen and select the papers, extract the data and assess the risk of bias. Apart from exploring the trends and types of CADD methods used, the target proteins of these compounds in cancer treatment will also be revealed in this review. It is believed that the outcome of this study could be utilized to support the ongoing research in similar area with better quality and greater probability of success, consequently optimizing the resources in subsequent *in vitro*, *in vivo*, non-clinical and clinical development. It will also serve as an evidence based scientific guide for new research to design novel anthraquinone-derived drug with improved efficacy and safety profile for cancer treatment.

**Competing interests:** The authors have declared that no competing interests exist.

## Introduction

Cancer remains one of the leading causes of deaths globally. The worldwide cancer burden is constantly on the rise and in estimation, the number of cancer cases will reach 28.4 million in year 2040 as compared to 19.3 million cases in year 2020 [1]. Cancer treatments typically involve a multidisciplinary approach, which may include surgical oncologist, radiation oncologist, medical oncologist and other specialists depending on the individual's case [2]. It is worth noting that drug resistance related to chemotherapy, radiotherapy or immunotherapy is a common issue that limits treatment efficacy in cancer patients, as well as the treatment-associated serious adverse events that raised safety concern of many of the current anti-cancer regimens [3]. Therefore, the search for new drugs with better efficacy and lesser side effects is always under the spotlight of researchers and pharmaceutical industries. However, the process of drug discovery and development is well known to be complex, lengthy and costly. This process can take up to 15 years [4], whereas the research and development (R&D) cost of a new drug is estimated to be USD $2.8 billion based on a survey conducted in year 2016 [5].

In recent years, technology advancement and computer power enhancement has enabled the utilization of various *in silico* tools to facilitate this process [6]. The term "*in silico*" derived from Latin phrase which means "in silicon", alluding to the use of silicon computer chips in computer technology. Compared to the traditional wet-lab experiments conducted in a laboratory setting (*in vitro*) or testing performed in living organisms (*in vivo*), *in silico* rely on computer-based algorithms, simulations, and modelling to expedite and optimize the drug design and discovery process [7].

Computer-aided drug design (CADD) is a broader term widely used to illustrate the application of computational techniques and approaches in designing new therapeutics via a rational and systematic manner. CADD has gained popularity and shown to be able to dramatically cut down the time and resources required especially in the early stages of the drug discovery and development pipeline [8]. These include computational identification of potential drug targets, virtual screening of chemical libraries to identify potential hit, applying *in silico* filter to discard molecules with poor Absorption, Distribution, Metabolism, Excretion properties and undesirable Toxicity (ADMET), shortlisting of lead (most likely drug candidate) for further evaluation and optimization [9]. Popular CADD tools include molecular docking, molecular dynamic simulation, quantitative structure-activity relationship (QSAR), similarity search, pharmacophore mapping and scaffold hopping [10]. De novo drug design is another significant *in silico* approach that enables the design of novel drug from scratch guided by the target binding site or pharmacophore model [11].

In virtual screening, potentially active molecules are searched in either readily available chemical space or from combinatorial libraries created computationally [12]. The ultimate goal of virtual screening is to reduce the size of chemical space to a manageable subset so that resources can be focused on the most promising candidates to be synthesized and tested in the laboratory [13]. This is not an easy task since the number of chemically accessible compounds that exist in the chemical space is almost equal to infinity based on current science and technology capabilities. Furthermore, there are still many unexplored regions which may contain novel drug candidates for newly identified target. Molecules derived from natural sources such as plants, microorganisms or aquatic species contain unique scaffold that can be utilized to construct natural product-like combinatorial libraries with great diversity and potential for discovering novel chemotype [14].

Natural products contain many biologically active substances or phytochemicals that carry medicinal value for treatment of various diseases including cancer [15]. Considerable amount of research effort has been made in the past decades to isolate or design novel natural products

**Fig 1. Naturally occurring anthraquinone-derivatives and their chemical structures depiction.**

in oncology setting as well as other disciplines. A review published in year 2020 estimated that out of a total of 185 small molecules authorized for cancer treatment in between year 1981 and 2019, only 16% are purely synthetic and the rest are either naturally derived (34%) or somehow inspired by the properties or scaffold of natural products (51%) [16].

Anthraquinones are a group of naturally occurring compounds that can be found in a variety of plants like aloe, rhubarb, and buckthorn, as well as some bacteria, fungi and animal [17]. The common examples of naturally occurring anthraquinone-derivatives include emodin, aloe emodin, rhein, chrysophanol, physcion, and alizarin (Fig 1). They exhibit a wide range of biological activities, including anti-cancer, anti-inflammatory, antibacterial, antiviral, antifungal, antimalaria, antidiabetic, antifibrotic, neuroprotective and laxatives effect [18,19].

The core structure of anthraquinone is an anthracene ring (a tricyclic aromatic ring) with carbonyl groups mostly at position 9 and 10. The 9,10-anthraquinone moiety are privileged chemical scaffolds that serve as the valuable starting point for design and development of anthraquinone analogues with a variety of pharmaceutical properties [20]. Chemical drugs that contain anthraquinone scaffold such as anthracycline (daunorubicin, doxorubicin, idarubicin, epirubicin) and synthetic anthraquinone like mitoxantrone and pixantrone are already available in the clinic to treat different types of cancer [21].

While there have been many extensive reviews discussing the medicinal role and function of anthraquinone or anthraquinone-derivatives in various disciplines [18–20,22–24], none of them are systematic review. There is one paper titled as "systematic review" summarizing the discovery and development of novel anthraquinone analogues for the treatment of various cancers in between year 2005 to 2021 [21], however there is no method section to explain the article searching and selection process, and in overall the format of the paper did not follow the PRISMA reporting guideline for systematic review [25]. Apart from that, there is also no review found which focus on computer aided drug design and discovery based on anthraquinone scaffold for cancer treatment. One of our recent paper comprehensively reviewed the power of computer tools and algorithm in drug discovery and lead optimization of

macrocyclic compounds such as anthraquinone derivatives to cater for different medical needs [26], however the emphasize was more on artificial intelligence approach and machine learning tools, and again, it is a narrative review instead of a systematic review.

This protocol aims to guide the first systematic review synthesizing the evidence of computer-aided drug design and discovery based on anthraquinone scaffold for cancer treatment. The objectives are to analyse the trends and types of CADD or virtual screening methods together with the software and database of choice used, as well as to gain further insights into the target proteins and therapeutic potential of anthraquinone analogues in different types of cancer.

## Methods/Design

### Protocol registration

This systematic review protocol was designed according to the recommendations of the "Preferred reporting items for systematic reviews and meta-analysis protocols (PRISMA-P) 2015" [27,28]. It was published in the "International prospective register of systematic reviews (PROSPERO)" database, (registration number CRD42023432904), accessible via: https://www.crd.york.ac.uk/prospero/display_record.php?ID=CRD42023432904.

### Review question

The protocol was conducted to answer the main research questions as follows:

"What are the trends and types of computer-aided drug design and discovery tools used in virtual screening based on anthraquinone scaffold for cancer treatment"?

"What are the therapeutic potential and target protein of anthraquinone and derivatives elucidated by CADD to treat cancer?

The questions were established according to the PECo strategy (P, problem; E, exposure; Co, context) for systematic review as outlined in Table 1.

### Eligibility criteria

**Inclusion criteria. Problem**: Studies with the clear description of the CADD tools used in virtual screening based on anthraquinone scaffold will be included.

**Exposure**: Studies investigating therapeutic potential and target protein of compounds containing anthraquinone scaffold for cancer treatment will be included.

**Context**: Only original research studies utilizing CADD techniques or virtual screening tools for the purpose of either target protein prediction/ validation, hit identification, hit-to-lead and lead optimization will be included.

**Table 1. PECo strategy for the systematic review.**

| Element | Abbreviation | Description |
|---------|--------------|-------------|
| Problem | P | Trends and types of computer-aided drug design and discovery methods used in virtual screening as primary tool to discover or design anticancer drug based on anthraquinone scaffold. |
| Exposure | E | Cancer and the therapeutic targets involved. |
| Context | Co | Original research studies with computer-aided drug design and discovery methods used in virtual screening as primary tool to serve the purpose of either target protein identification/ validation, hit identification, hit-to-lead or lead optimization. |

**Exclusion criteria. Problem**: Studies without details or clear description of the CADD or virtual screening tools will be excluded.

Studies not involving compounds with anthraquinone scaffold will be excluded.

**Exposure**: Studies investigating diseases other than cancer and involving non-human target protein will be excluded.

**Context**: Studies exclusively *in vivo*, *in vitro* and other types of *in silico* tools that do not serve the purpose of target protein prediction/validation, hit identification, hit-to-lead or lead optimization will be excluded. Network pharmacology is beyond the scope of this review and will be excluded. Review article, book chapter, letters, grey literatures (conference paper abstracts, theses/ dissertation, report) will be excluded.

## Information sources

The following electronic databases will be searched: PubMed (https://pubmed.ncbi.nlm.nih.gov/), Scopus (https://www.scopus.com/), Web of Science (https://www.webofscience.com/) and MedRxiv (https://www.medrxiv.org/).

## Search strategy

The search strategy aims to include studies published in English that used CADD tools like molecular docking or molecular dynamic simulation or any other virtual screening method in the search for anticancer drug containing anthraquinone scaffold. There will be no restriction on the publication period. The search equation for the systematic review will be defined considering the items of the PECo strategy (Table 1). Comprehensive search will be performed using medical subject headings (MeSH) and Boolean operators (AND and OR). The search strategy may be adjusted accordingly based on the different characteristics of the electronic databases. The following main search terms will be used with focus on article title, abstract and keywords (Example of Scopus database):

("virtual screening" OR "computer aided drug design" OR "molecular docking" OR "molecular dynamics") AND ("anthraquinone" OR "anthracenedione" OR "anthranoid" OR "anthradione" OR "dioxoanthracene" OR "anthracene-9,10-dione" OR "anthracene-9,10-quinone" OR "9,10-anthrachinon" OR "9,10-dihydro-9,10-dioxoanthracene") AND (Cancer OR tumour OR malignant OR neoplasm).

## Study selection

Endnote X9.0 will be used to manage the retrieved studies and remove the duplicates. After deduplication, the titles and abstract of the studies will be screened by two independent reviewers to identify articles that potentially meet the inclusion criteria. The full-text articles of selected studies will be retrieved and read in detail by two reviewers separately to further assess the eligibility. The reasons for exclusion will be recorded. Any disagreement between the two reviewers will be resolved through discussion with a third reviewer. PRISMA-2020 flowchart (Fig 2) will be used to record and report the study selection process [25].

## Risk of bias/quality assessment

Due to the lack of standardised tool for this type of study, the quality and risk of bias of the selected papers which involve molecular docking will be assessed by adopting a checklist previously developed and applied by Taldaev et al [29]. The assessment will be carried out separately by two independent reviewers. Any discrepancies will be resolved by a third reviewer. Training will be carried out with the reviewers to ensure uniformity in applying the checklist.

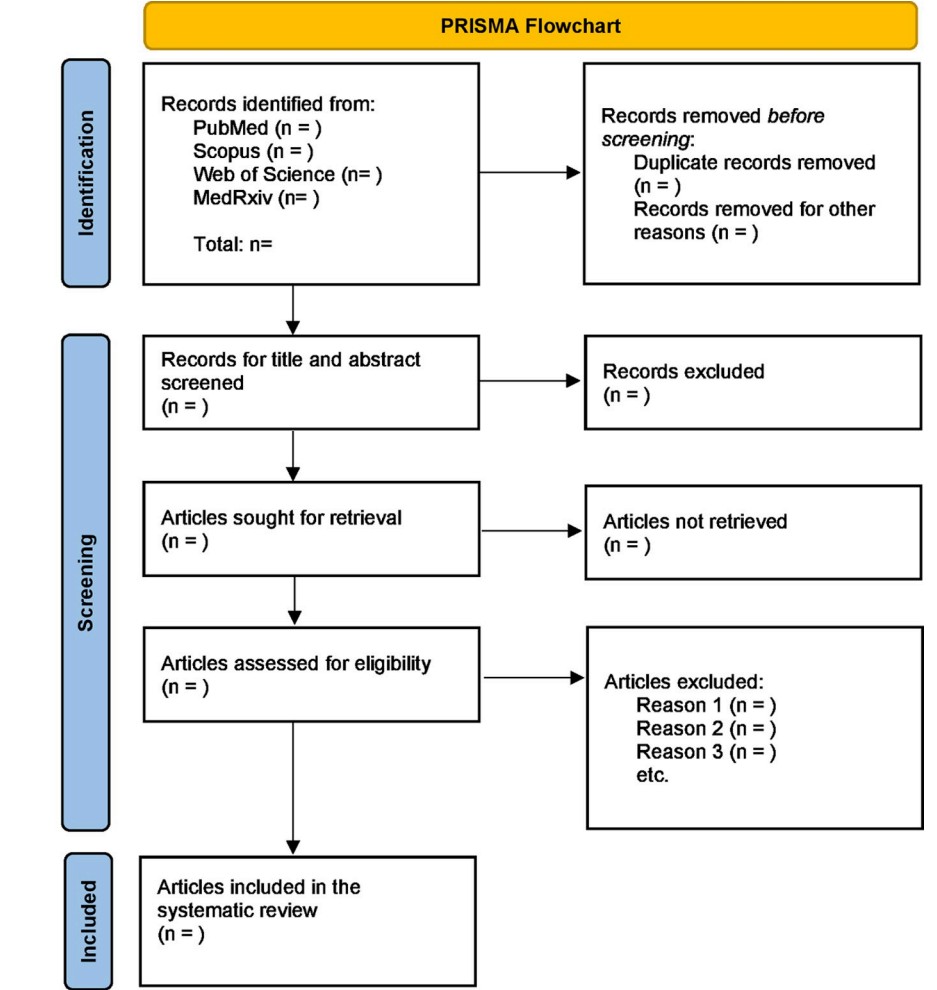

**Fig 2. Study selection flowchart (adapted from PRISMA-2020 flow diagram).**

## Data extraction

Two independent reviewers will extract data from the eligible studies using a predefined data extraction form. The characteristics of the included studies to be extracted including title of journal, authors, publication year, CADD methods used and their description, the molecules/compounds investigated, macromolecular targets and medical conditions (for example for general cancer or specific cancer type) involved, software and database used, and other relevant data.

## Data synthesis and analysis

A narrative approach will be used to summarize the data. Tables and figures will be used to present the characteristics of the studies. The trends and types of different computer-aided drug design and discovery methods used in virtual screening as primary tool for designing or discovering anticancer drug based on anthraquinone scaffold will be analyzed and summarized. The respective macromolecular targets involved will be also identified and discussed.

## Reporting

The "PRISMA 2020 statement: an updated guideline for reporting systematic reviews" will be followed for the reporting of the systematic review [25].

### Ethics and dissemination plans

Given that there will be no patients recruited, ethical approval is not required for the conduct of this review. The results of this review will be disseminated in a peer-review journal.

## Discussion

With the arising number of cancer cases worldwide that put many lives under threat, the scientific and industry effort in discovering better treatment option is also gaining momentum especially by exploring the nature [30]. Natural products and their derivatives are rich reservoir for drug discovery as they have survived over millions of years of evolution by producing secondary metabolites with diverse structures to endure the environmental challenges. For example, paclitaxel is one of the most well-known anticancer drugs isolated from plant, discovered since many decades ago and it's still being used in the clinic today [31].

Anthraquinone, a natural compound with planar tricyclic aromatic system has attracted interest of many researchers due to its privileged scaffold that carry a broad spectrum of biological activities including anticancer properties. Over 75 naturally occurring anthraquinones have been isolated from medicinal plants, algae, fungi and marine reservoir. [23]. The antitumour activities of anthraquinone are mediated via various mechanisms which include DNA damage of cancer cells, cell cycle arrest, apoptosis, autophagy, as well as alteration of key signalling pathway involved in the physiological process. Due to these unique properties, naturally derived 9,10-anthraquinone such as rhein, emodin and aloe emodin have been commonly used by researchers as starting points in the design of anticancer drugs for the past few decades [20].

Before a drug can be marketed, it undergoes a complicated process starting from research and development, preclinical testing on cell-based and animal models, followed by trials on human subjects [32]. The attrition rate of drug discovery project is high and in estimation, only 10 out of ten thousand synthesized and tested compounds managed to enter clinical trials, where only 1 candidate passed through regulatory assessment and licensed for medical use [33].

By making use of computational software, the process can be streamlined to a more cost-effective manner especially in the early phases of the drug discovery pipeline, which include target protein identification and validation, hit identification, hit-to-lead and lead optimization. Computer-aided drug design (CADD) tools in virtual screening help researchers to prioritise the most promising candidates, reduce the time and cost needed to perform testing on large batches of compounds in the laboratories, subsequently reducing the use of animal models as well as increase the success rate of clinical trial [34].

The utilization of computer-aided approaches to develop novel drug possessing anthraquinones moieties for cancer treatment are getting more intense [23]. Since CADD approaches play significant role in modern drug design and discovery trajectory, it is crucial to examine the different types of tools available together with their prospect, limitations and challenges [32].

There are some potential limitations of this review. The research papers found and included in the review may be of low quality, or the CADD methods used in the studies may be lack of details description. Heterogenicity of the studies recruited due to different hardware, software or algorithm used may contribute to the review limitation as well.

Nevertheless, it is believed that the outcome of this study could be utilized to support the research that is already in progress in the area of computer-aided drug design and discovery based on anthraquinone scaffold with better quality and greater probability of success, consequently optimizing the resources in subsequent *in vitro*, *in vivo*, non-clinical and clinical

development. It will also serve as an evidence based scientific guide for new research using *in silico* methods to design novel anthraquinone-derived drug with better efficacy and safety profile that can fulfil the unmet medical needs.

## Supporting information

**S1 Checklist. S1 PRISMA-P 2015 checklist.**
(PDF)

## Acknowledgments

We thank Universiti Brunei Darussalam for the University Graduate Scholarship awarded to HMC and SM.

## Author Contributions

**Conceptualization:** Long Chiau Ming, Nurolaini Kifli.

**Data curation:** Hui Ming Chua, Said Moshawih.

**Formal analysis:** Hui Ming Chua, Said Moshawih.

**Investigation:** Hui Ming Chua, Said Moshawih.

**Methodology:** Hui Ming Chua, Said Moshawih, Hui Poh Goh, Long Chiau Ming, Nurolaini Kifli.

**Project administration:** Nurolaini Kifli.

**Supervision:** Nurolaini Kifli.

**Validation:** Long Chiau Ming.

**Writing – original draft:** Hui Ming Chua.

**Writing – review & editing:** Hui Ming Chua, Said Moshawih, Hui Poh Goh, Long Chiau Ming, Nurolaini Kifli.

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
