## [Decision Letter · Decision Letter 0]

9 Aug 2023

PONE-D-23-19138Insights into the computer aided drug design and discovery based on naturally occurring 9,10-anthraquinone scaffold for cancer treatment: A protocol for systematic reviewPLOS ONE

Dear Dr. Chua,

Thank you for submitting your manuscript to PLOS ONE. After careful consideration, we feel that it has merit but does not fully meet PLOS ONE’s publication criteria as it currently stands. Therefore, we invite you to submit a revised version of the manuscript that addresses the points raised during the review process.

ACADEMIC EDITOR: Reading the title of the protocol together with the body text, it is not clear whether the focus of the study will be specifically on the 9,10-anthraquinone, or generally on the anthraquinones and anthraquinone-like scaffolds. This issue should come out clearly. It would be prudent to depict the chemical structures of some of the naturally occurring compounds that contain the 9,10-anthraquinone scaffold. All typographical errors, especially those concerning the tenses should be addressed as appropriate.==============================

We look forward to receiving your revised manuscript.

Kind regards,

Peter Mbugua Njogu, Ph.D.

Academic Editor

PLOS ONE

Reviewers' comments:

Reviewer's Responses to Questions

**Comments to the Author**

1. Does the manuscript provide a valid rationale for the proposed study, with clearly identified and justified research questions?

Reviewer #1: Yes

Reviewer #2: Yes

2. Is the protocol technically sound and planned in a manner that will lead to a meaningful outcome and allow testing the stated hypotheses?

Reviewer #1: Partly

Reviewer #2: Yes

3. Is the methodology feasible and described in sufficient detail to allow the work to be replicable?

Reviewer #1: Yes

Reviewer #2: Yes

4. Have the authors described where all data underlying the findings will be made available when the study is complete?

Reviewer #1: Yes

Reviewer #2: Yes

5. Is the manuscript presented in an intelligible fashion and written in standard English?

Reviewer #1: Yes

Reviewer #2: Yes

6. Review Comments to the Author

You may also provide optional suggestions and comments to authors that they might find helpful in planning their study.

Reviewer #1: The Search strategy should clearly be in tandem with the impression created by the title of the manuscript.

Reading the title of the protocol together with the body text, it is not clear whether the focus of the study will be specifically on the 9,10-anthraquinone, or generally on the anthraquinones and anthraquinone-like scaffolds. This issue should come out clearly.

For example, under the Search Strategy (Lines 207-221), some of the search terms will undoubtedly retrieve anthraquinone-like scaffolds, some of which will be chemically at variance from the intended 9,10-anthraquinone moiety. Will the proposed PRISMA-P flowchart exclude studies not reporting specifically on 9,10-anthraquinone scaffold?

Reviewer #2: Some very minor typographical errors e. g. line 86, word 'that' after approach should be inserted. Similarly, line 96, after the word compounds. Line 105, replace the word 'considerably' with 'considerable'.

7. PLOS authors have the option to publish the peer review history of their article (what does this mean?). If published, this will include your full peer review and any attached files.

Reviewer #1: No

Reviewer #2: No

---

## [Author Response · Author response to Decision Letter 0]

18 Aug 2023

Comments to Reviewer 1: 

We appreciate the reviewer's observation regarding the potential ambiguity between the 9,10-anthraquinone scaffold and other anthraquinone-like scaffolds. 

To clarify, the primary focus of our study is on anthraquinone, in this case it may include 9,10-anthraquinone, anthraquinone in general or even anthraquinone-like scaffold. In addition, all studies that utilized CADD tools in designing or discovering anticancer drugs containing anthraquinone scaffold, either naturally derived, semi-synthetic or synthetic will be also included in the review. Therefore, the search strategy (Lines 207-221) was developed to retrieve all possible studies in a comprehensive manner by including several different keywords/synonyms for anthraquinone.

In such case, we have revised the title to: "Insights into computer-aided drug design and discovery based on anthraquinone scaffold for cancer treatment: A protocol for systematic review." 

Comments to Reviewer 2:

We apologize for the oversight and have thoroughly reviewed the manuscript for typographical and grammatical errors. All instances as pointed out, from Line 25 to Line 303, have been corrected in line with the recommendations.

We have also ensured consistency in terminology for PRISMA as suggested.

---

## [Editor Report · Decision Letter 1]

21 Aug 2023

Insights into the computer-aided drug design and discovery based on anthraquinone scaffold for cancer treatment: A protocol for systematic review

PONE-D-23-19138R1

Dear Dr. Chua,

We’re pleased to inform you that your manuscript has been judged scientifically suitable for publication and will be formally accepted for publication once it meets all outstanding technical requirements.

Kind regards,

Peter Mbugua Njogu, Ph.D.

Academic Editor

PLOS ONE

---

## [Editor Report · Acceptance letter]

25 Aug 2023

PONE-D-23-19138R1 

Insights into the computer-aided drug design and discovery based on anthraquinone scaffold for cancer treatment: A protocol for systematic review 

Dear Dr. Chua:

I'm pleased to inform you that your manuscript has been deemed suitable for publication in PLOS ONE. Congratulations! Your manuscript is now with our production department. 

Kind regards, 

on behalf of

Dr. Peter Mbugua Njogu 

Academic Editor

PLOS ONE